# OpenReview forum: "Memory Makes The Poisons: Understanding and Mitigating Data Poisoning in LVLMs"
_TMLR — Under review for TMLR_

### Review · Reviewer_Cn5i · 2026-05-20

**Summary Of Contributions:**

The paper shows that data memorization, not just adversarial visual perturbations, is a major contributor to ShadowCast poisoning attack success on LVLMs, especially at higher poison ratios. They demonstrate this via controlled experiments removing perturbations while keeping setups identical, and show multimodal training exacerbates memorization relative to text-only. They propose RejectShield, a rejection-based defense using an adversarial image detector to filter poisoned samples before fine-tuning, achieving up to 99% attack success reduction.

**Audience:**

Yes

**Audience Explanation:**

The memorization observation is practically relevant for anyone fine-tuning LVLMs on crowd-sourced data. The finding that purification defenses are structurally insufficient at higher poison ratios is useful.

**Claims And Evidence:**

Yes

**Claims Explanation:**

The core ablation (removing perturbations, observing high attack success persists) is well-designed and the results are consistent across models and tasks. However, there are three substantive issues:
(1) Defense doesn't address the identified root cause. The paper's thesis is that memorization, not perturbation, is the primary driver at moderate-to-high poison ratios. But RejectShield detects adversarial perturbations in images, so it has nothing to do with memorization. If an attacker injects clean (unperturbed) destination-concept images with matching captions, RejectShield is blind to it, yet the paper's own controlled experiments show this achieves >90% attack success at >=2% poison ratio. This is a fundamental disconnect between the analysis contribution and the defense contribution. The paper needs to either reframe the defense as targeting ShadowCast's perturbation pipeline specifically (not the memorization mechanism), or propose a defense that actually addresses memorization.
(2) No adaptive attack evaluation. RejectShield relies on a fixed adversarial detector. The paper does not evaluate against an adversary who knows about the detector and crafts perturbations to evade it. This is a standard requirement for defense papers in adversarial ML (Tramer et al., 2020). The JPEG-augmented and LAVIS-augmented variants are not adaptive, they don't target the detector.
(3) LVLM-vs-LLM comparison has uncontrolled confounds. The two settings use different fine-tuning datasets (Sub-CC-Aligned vs. Sub-Alpaca) with different data distributions. The injected content also differs in format. Attributing the entire gap to "multimodality" is not warranted when the textual distributions are not matched. This weakens Finding 2.

**Requested Changes:**

(1) Explicitly address the mismatch between the analysis (memorization is the root cause) and the defense (detects perturbations, not memorization). Either reframe the narrative or demonstrate defense effectiveness against clean-image poisoning (i.e., the "no perturbation" variant that the paper itself shows is effective).
(2) Evaluate against adaptive attacks where the adversary optimizes perturbations to evade the specific detector used in RejectShield.
(3) Tighten the LVLM-vs-LLM comparison by controlling for textual data distribution, e.g., fine-tune the LLM on the text component of Sub-CC-Aligned rather than Sub-Alpaca, so the only variable is image presence.
(4) Discuss feasibility of the high poison ratio regime. At 5% of a 3.5k dataset, ~175 samples share the same concept; this seems detectable by trivial deduplication. The practically interesting regime is low ratios, where the memorization story is weakest.

---

> ### Author Response · Authors · 2026-06-07
> **Authors' Reply: Thank You for the Thoughtful and Supportive Feedback**
>
> We sincerely thank the reviewer for your time and valuable comments.
>
> In what follows, we provide detailed responses to all requested changes from the reviewer. The revised PDF will be submitted right after 3 reviews are submitted per TMLR rebuttal guideline suggestion.
>
> > **Request 1:** "Explicitly address the mismatch between the analysis (memorization is the root cause) and the defense (detects perturbations, not memorization). Either reframe the narrative or demonstrate defense effectiveness against clean-image poisoning (i.e., the "no perturbation" variant that the paper itself shows is effective)"
>
> We agree with the reviewer and reframe the narrative that RejectShield targets the ShadowCast-like threat model, not all memorization-based poisoning.
>
> To address the reviewer’s broader concern, we additionally provide preliminary evidence toward memorization-aware mitigation in Appx. B6. We frame this as preliminary evidence and a direction for stronger future defenses.
>
> > **Request 2:** "Evaluate against adaptive attacks where the adversary optimizes perturbations to evade the specific detector used in RejectShield."
>
> We thank the reviewer for the suggestion. Following the reviewer’s recommendation, we implemented an adaptive variant that explicitly optimizes perturbations to evade the RejectShield detector during poisoning, where the poisoning objective is as below, where $\mathcal{L}_R$ denotes the RejectShield detector loss.
>
> $$
> \min_\delta \|\phi(x_d + \delta) - \phi(x_o)\|_2^2 + \lambda \mathcal{L}_R (x_d + \delta)
> $$
>
> This is a very challenging defense setup since the assumption is that attacker has full access to the defense detector. We observe that the adaptive attack significantly weakens the detector’s effectiveness. We will include this result and clarify the scope of RejectShield in the revision that RejectShield is effective against standard ShadowCast-style perturbation attacks, particularly in the low-poison regime, but is not a fully robust defense against adaptive adversaries. This result further motivates future memorization-aware defenses, for which we provide preliminary evidence in Appx. B6.
>
> | Poison Ratio (%) | ASR ShadowCast | ASR JPEG-augmented | ASR LAVIS-augmented | ASR Adaptive |
> |------------------|---------------|-------------------|--------------------|-------------|
> | 0.0 | 0.0 | 0.0 | 0.0 | 0.0 |
> | 0.1 | 0.0 | 0.0 | 0.0 | 1.5 |
> | 0.3 | 0.5 | 0.0 | 0.0 | 8.5 |
> | 0.6 | 0.0 | 0.0 | 0.0 | 61.0 |
> | 0.9 | 0.0 | 0.0 | 0.0 | 79.5 |
> | 1.4 | 0.0 | 0.0 | 0.0 | 93.0 |
> | 2.9 | 0.0 | 0.0 | 0.0 | 97.5 |
> | 4.3 | 16.5 | 0.0 | 0.0 | 96.5 |
> | 5.7 | 8.5 | 0.0 | 0.0 | 98.0 |
>
> > **Request 3:** "Tighten the LVLM-vs-LLM comparison by controlling for textual data distribution, e.g., fine-tune the LLM on the text component of Sub-CC-Aligned rather than Sub-Alpaca, so the only variable is image presence."
>
> We thank the reviewer for this valuable suggestion. Following the reviewer’s recommendation, we conduct the same LVLM-vs-LLM comparison while fine-tuning the LLM on the text component of Sub-CC-Aligned instead of Sub-Alpaca. **The results below remain consistent with our observation in Sec. 3.3 that under comparable textual data, the LVLM setup still exhibits much stronger memorization behavior than the LLM-only setup.**
>
> | Poison Ratio (%) | ASR - LLM Setup | ASR - LVLM Setup |
> |------------------|----------------|------------------|
> | 0.0 | 5.24 | 0.00 |
> | 0.1 | 5.24 | 0.00 |
> | 0.3 | 7.33 | 1.50 |
> | 0.6 | 5.24 | 4.00 |
> | 0.9 | 6.28 | 20.50 |
> | 1.4 | 6.81 | 44.00 |
> | 2.9 | 7.85 | 93.50 |
> | 4.3 | 9.42 | 93.50 |
> | 5.7 | 8.90 | 97.00 |
>
> > **Request 4:** "Discuss feasibility of the high poison ratio regime. At 5% of a 3.5k dataset, ~175 samples share the same concept; this seems detectable by trivial deduplication. The practically interesting regime is low ratios, where the memorization story is weakest."
>
> We agree that high poison ratios could be easier to detect. However, for clarification, our main claim is not that memorization only exists at a 5% poison ratio. The memorization phenomenon is still observable at lower poison ratios (e.g., at 1.4%, the ASR still reaches around 60% for the Engine-to-Fuel task without visual perturbations).
>
> Importantly, RejectShield remains particularly effective in the practically relevant low-poison regime. Our goal in studying higher poison ratios follows ShadowCast attack setups and helps better understand the transition between perturbation-driven and memorization-driven attack behavior. We clarify this distinction in the revision and better separate practical low-poison regimes from high-ratio stress-test regimes.

---

### Review · Reviewer_EpHv · 2026-06-04

**Summary Of Contributions:**

This manuscript analyze the ShadowCast data poisoning method on LLM and LVLM with the same LLM backbone and find that visual + text poison data for LVLM could heavily impact the model performance while the text poison data for LLM have less impact (Fig. 3). The author(s) then proposed the "RejectShield" method that filters poison image text pairs int fine-tuning dataset to avoid such effect. The method outperform image purification methods on LLaVA NeXT and MiniGPT4-v2 models.

**Audience:**

Yes

**Audience Explanation:**

Data poisoning defense is of important for LLM and LVLM applications.

**Claims And Evidence:**

Yes

**Claims Explanation:**

The controlled experiment of using the same LLM backbone and compare data poisoning method  ShadowCast  on LLM and LVLM shows convincing evidence of the argument that the data poisoning effect  is determined largely by the multi-modal setting.

**Requested Changes:**

1. Please provide more details on how the adversarial detector is designed and trained, which could strengthen the contribution.
2. (optional) For completeness, it would be better if visual-only experiment could be provided in Fig. 3.

---

> ### Author Response · Authors · 2026-06-07
> **Authors' Reply: Thank You for the Thoughtful and Supportive Feedback**
>
> We sincerely thank the reviewer for your time and valuable comments.
>
> In what follows, we provide detailed responses to all requested changes from the reviewer. The revised PDF will be submitted right after 3 reviews are submitted per TMLR rebuttal guideline.
>
> > **Request 1**: "Please provide more details on how the adversarial detector is designed and trained, which could strengthen the contribution."
>
> We thank the reviewer for this suggestion. We agree that providing more implementation details on the detector would strengthen the presentation, and we revise the paper accordingly.
>
> Concretely, RejectShield is a binary image classifier with an Xception backbone followed by a cosine-similarity head. The backbone produces a 2048-dimensional visual feature representation. This feature is then fed to a cosine-similarity head rather than a standard affine MLP head that instead of computing logits as (Wx+b), the model compares the normalized feature vector against learned class prototype vectors for the clean and adversarial classes. This design encourages the detector to rely primarily on angular separation between clean and adversarial features.
>
> For training, we follow Wang et al. (2024) and further fine-tune the pretrained detector on 500 randomly sampled COCO images together with their adversarial counterparts, using the feature-transfer strategy of Huang et al. (2019). In our implementation, optimization is performed with cross-entropy loss for 10 epochs using Adam, with learning rate $2 \times 10^{-4}$ and weight decay $5 \times 10^{-6}$. We select the final checkpoint based on validation AUC.
>
> At inference time, the detector outputs a clean-score probability, and we use a fixed operating threshold selected based on the TPR/FPR trade-off. Importantly, detector training is lightweight and does not require any ShadowCast-poisoned samples, and the same threshold is used across all experimental setups. We already provide additional threshold analysis and detector ablations in the supplementary material (Fig. 14 and Tab. 5). In the revision, we make these references explicit in the main text and add the above engineering details to clarify both the detector architecture and training procedure.
>
> > **Request 2**: "(Optional) For completeness, it would be better if a visual-only experiment could be provided in Fig. 3."
>
> We thank the reviewer for this suggestion. We conduct a visual-only experiment corresponding to Fig. 3 and will include the results in the revision.
>
> For clarification, the goal of Fig. 3 is to isolate the effect of multimodal inputs under a controlled architecture comparison. Specifically, the LLM-only setup uses language-only input, while the LVLM setup uses both language and visual input. All other components, including the Vicuna-1.5-7B backbone and fine-tuning procedure, remain comparable. A fully vision-only model (i.e., an image-to-text model) would require a different architecture and decoding head, making it less directly comparable to the current LLM-versus-LVLM setup.
>
> To maintain the same LVLM architecture and training pipeline while varying only the conditioning modality, the closest vision-only control is to retain the LVLM setup but remove textual information from the injected samples so that images become the sole conditioning signal. Specifically, each injected sample is transformed from `(image, text prompt, target response)` to `(image, null prompt, target response)`, while keeping the image and target response unchanged.
>
> The results are shown below. The vision-only control confirms that visual input alone can induce the same hallucination threat, but it is significantly less effective than the full LVLM setup at low-to-moderate poison ratios. This suggests that visual supervision contributes materially to the threat, while the full multimodal setup remains the most sample-efficient regime for inducing targeted memorization.
>
> | Poison Ratio (%) | ASR - LLM Setup (Language-only Input) (%) | ASR - LVLM Setup (Vision-Language Input) (%) | ASR - LVLM Setup (Vision-only Input) (%) |
> |-|-|-|-|
> |0.0|5.24|0.00|0.00|
> |0.1|3.66|0.00|0.00|
> |0.3|4.71|1.50|0.50|
> |0.6|6.28|4.00|2.50|
> |0.9|3.66|20.50|6.50|
> |1.4|4.71|44.00|13.00|
> |2.9|2.62|93.50|84.50|
> |4.3|2.62|93.50|93.00|
> |5.7|3.14|97.00|94.50|
>
> #### **References**
> Qian Wang, Chen Li, Yuchen Luo, Hefei Ling, Shijuan Huang, Ruoxi Jia, and Ning Yu. Detecting adversarial data using perturbation forgery. arXiv preprint arXiv:2405.16226, 2024.
>
> Qian Huang, Isay Katsman, Horace He, Zeqi Gu, Serge Belongie, and Ser-Nam Lim. Enhancing adversarial example transferability with an intermediate level attack. In Proceedings of the IEEE/CVF international conference on computer vision, pages 4733–4742, 2019.

---

### Review · Reviewer_v8Gk · 2026-06-16

**Summary Of Contributions:**

The paper revisits the ShadowCast data poisoning attack against large vision-language models. Its main contribution is a re-analysis of the mechanism behind ShadowCast’s success. Whereas prior work attributed the attack primarily to adversarial visual perturbations, the authors argue that memorization during LVLM fine-tuning is also a major contributor, especially at moderate and high poison ratios. They support this claim through controlled experiments in which the visual perturbations are removed while poisoned image-caption associations are retained; attack success remains high in several settings. The paper further compares multimodal LVLM fine-tuning with a text-only LLM setting and argues that multimodal inputs exacerbate memorization-driven vulnerability. Based on this diagnosis, the authors propose RejectShield, a rejection-based defense that filters suspicious training samples before fine-tuning using an adversarial-image detector. The defense is evaluated across several attack goals, LVLM architectures, black-box and white-box settings, and augmented ShadowCast variants, with reported reductions in attack success while largely preserving standard benchmark utility.

**Additional Comments:**

NAN

**Audience:**

No

**Audience Explanation:**

See the report above.

**Broader Impact Concerns:**

Not needed.

**Claims And Evidence:**

No

**Claims Explanation:**

\section*{Overall Assessment}

This is a clearcut rejection.

The paper contains a promising empirical observation, but the current version overstates its conclusions and does not adequately validate the proposed defense against the threat model implied by its own analysis.  The authors argue that memorization is the (overlooked?) driver of ShadowCast's success. However, RejectShield is presented as an adversarial-image detector. This implies the defense primarily targets the perturbation-based component of the attack, not the memorization mechanism that the paper claims is central.

The authors' own experiments show that removing visual perturbations does not remove the vulnerability at moderate and high poison ratios. Therefore, the natural and necessary evaluation is to test RejectShield against clean or unperturbed poisons that still induce memorization. This experiment appears essential -- however, it is not presented as a central result. Without it, the paper has not shown that RejectShield mitigates the memorization-driven failure mode. It has mostly shown that an adversarial-image detector can filter adversarially perturbed ShadowCast samples.

This substantially weakens the paper's main claim.

\section*{Strengths}

\begin{enumerate}[leftmargin=*]
\item The problem presented in the paper is relevant.


\item The re-analysis of ShadowCast is valuable.

\item The empirical results suggest that repeated poisoned associations during fine-tuning can induce targeted hallucinations.

\item The paper is generally well organized and the experimental narrative is easy to follow.


\end{enumerate}

\section*{Major Concerns}

\subsection*{1. The proposed defense does not address the paper's central mechanism}

This is the main reason for my rejection recommendation.

The paper argues that memorization is a major contributor to ShadowCast success and that perturbation-only defenses are insufficient. However, RejectShield is based on detecting adversarially manipulated images. This creates a direct conceptual mismatch: the paper motivates the defense by saying that visual purification is insufficient because the problem is memorization, but the proposed defense still relies on identifying visual adversarial manipulation.

If the attack can succeed without visual perturbations, as the paper claims, then an adversarial-image detector should not be expected to reject those clean poisoned samples. The most important missing experiment is therefore straightforward:
Does RejectShield work against the no-visual-perturbation poisoning setting?

If the answer is no, then RejectShield does not defend against the memorization vulnerability identified by the paper. If the answer is yes, the paper needs to explain why an adversarial-image detector rejects clean or unperturbed memorization poisons. Without this experiment, the defense contribution is not convincing.

\subsection*{2. The paper overclaims the interpretation of {\em memorization}}

The paper repeatedly states that data memorization is the major contributor or root cause of the attack. However, I do not see this as a direct evidence. High attack success without visual perturbations is consistent with memorization, but it is also consistent with other mechanisms: target-label frequency bias, shortcut learning, induced semantic prior shift, repeated-caption effects, class imbalance, or fine-tuning instability.

The paper needs direct memorization diagnostics. For example, the authors could test whether exact repetition matters, whether paraphrased target captions have the same effect, whether unique target examples produce the same result, whether nearest-neighbor or influence-function analyses trace outputs to injected samples, or whether the model assigns unusually high likelihood to memorized target captions. Without such evidence, ``memorization'' remains a plausible but insufficiently established explanation.

\subsection*{3. The defense is not evaluated against the natural adaptive attack}

The defense is not evaluated against the most immediate adaptive threat. Once the defender reveals that RejectShield relies on an adversarial-image detector, the attacker’s response is straightforward: avoid producing images that look adversarial to that detector. This could be done by using clean target-concept examples, very weak perturbations, detector-aware optimization, or poisons explicitly constrained to pass the detector. The paper does not convincingly evaluate any of these adaptive settings.

This omission is particularly serious because the paper’s own analysis shows why such attacks are plausible. If, at moderate and high poison ratios, visual perturbations are not necessary for high attack success, then an attacker need not rely on the very signal that RejectShield is designed to detect. In that regime, RejectShield may simply defend against non-adaptive, visibly perturbed ShadowCast-style samples, rather than against the memorization-driven vulnerability identified by the paper. This substantially weakens the claimed robustness of the defense.


\subsection*{4. The LVLM-vs-LLM comparison does not isolate multimodality}


\subsection*{5. The reported up to $99\%$ defense result is not sufficient}

The paper emphasizes large reductions in attack success, but the headline ``up to'' number is not enough to establish robustness. The paper should report averages across tasks, models, poison ratios, and seeds, together with variability. Fine-tuning large multimodal models with small poison ratios can be noisy, and the current presentation makes it difficult to assess whether the defense is consistently robust or selectively strong in certain settings.



\subsection*{6. The paper's claims about generalization are too broad}

The experiments cover several attack goals, models, and augmented ShadowCast variants. However, these are still close to the original attack family. The paper claims that RejectShield generalizes to unseen poisoning attacks, but the evidence does not support such a broad statement. Augmented ShadowCast poisonings are not the same as fundamentally different poisoning strategies, especially clean-label or detector-aware strategies.

**Requested Changes:**

I do not see how this paper could be improved and I do not know how to help the author(s) to make the paper competitive for TMLR.

---

> ### Author Response · Authors · 2026-06-25
> **Authors' Reply: Part 1/3**
>
> We sincerely thank the reviewer for your time and valuable comments.
> In what follows, we provide detailed responses to all requested changes from the reviewer. The revised PDF is updated to reflect our changes.
>
> > **Concern 1:** The proposed defense does not address the paper's central mechanism
>
> We thank the reviewer for this important comment and sincerely appreciate the opportunity to clarify two aspects of our submission. The first concerns the scope of the proposed defense: RejectShield is designed for the ShadowCast threat model studied in this paper. The second concerns the implications of our analysis: while we identify memorization as an important vulnerability underlying ShadowCast, this should not be interpreted as evidence that a practical, stealthy, and robust memorization-only poisoning attack already exists. In fact, this broader perturbation-free setting has already been partially explored in Sec. B.6. We clarify these points below.
>
> ***a. Scope of this work and the proposed defense.***
>
> Our primary contribution is a re-analysis of the ShadowCast attack, as stated in the abstract and correctly recognized by the reviewer. Specifically, we show that memorization is a major and previously overlooked contributor to ShadowCast's success. This finding reveals an important vulnerability in LVLM fine-tuning: attacker-specified associations can be memorized and later retrieved to induce targeted hallucinations. Motivated by this finding, we design RejectShield to defend against the ShadowCast threat model.
>
> ***b. Implications of our analysis.***
>
> While our analysis identifies memorization as a key vulnerability, we respectfully argue that discovering a vulnerability mechanism is different from establishing a practical attack that exploits that mechanism in isolation. In particular, identifying memorization as a contributor to ShadowCast does not imply that a practical, stealthy, and robust memorization-only poisoning attack already exists.
>
> To exploit this vulnerability directly, an adversary must still address several attack-design challenges, including maintaining stealth, avoiding simple data-quality and frequency-based monitoring, and achieving reliable attack success across different operating conditions. A straightforward approach is to repeatedly inject clean image-caption pairs containing the attacker-specified target concept, which is also the scenario raised by the reviewer. However, such attacks are comparatively easy to detect through concept-frequency monitoring. **In Sec. B.6, we show that repeated concept injection can be identified and mitigated by a simple monitoring mechanism (MemDefense), indicating that naive memorization-only attacks are not necessarily robust or stealthy.**
>
> Therefore, we view memorization as an important vulnerability revealed by our analysis rather than a fully developed attack paradigm. Developing perturbation-free poisoning attacks that exploit memorization while remaining difficult to detect is an important research direction, but falls beyond the scope of the present work.
>
> ***c. Clarification of RejectShield's scope.***
>
> We agree that the scope of RejectShield should be stated more clearly. RejectShield is designed to defend against the ShadowCast threat model, where poisoned samples are created through adversarially manipulated images. Our goal is not to claim a universal defense against all possible memorization-based poisoning attacks. **Rather, RejectShield is designed to prevent ShadowCast poison samples from entering the fine-tuning pipeline and subsequently being memorized by the model.**
>
> In the revised manuscript, we clarify that: (1) RejectShield is a defense for the ShadowCast threat model; (2) our analysis identifies memorization as a key vulnerability underlying ShadowCast, but does not establish the existence of a practical, stealthy, and robust perturbation-free memorization-only poisoning attack, consistent with the findings already reported in Sec. B.6; and (3) perturbation-free memorization-based poisoning represents a broader threat model that warrants future investigation.
>
> We thank the reviewer again for this important comment and appreciate the opportunity to provide these clarifications.

---

> ### Author Response · Authors · 2026-06-25
> **Authors' Reply: Part 2/3**
>
> > **Concern 2:** The paper overclaims the interpretation of memorization
>
> We thank the reviewer for this insightful comment. We agree that while high attack success without visual perturbations is consistent with memorization, direct memorization diagnostics are needed.
>
> Following Carlini et al. (USENIX Security 2019), we use **out-of-distribution canary tokens** as a diagnostic for memorization (see below for details of the experiments and the newly added supplementary section). Specifically, we insert unique canary tokens into poisoned training samples. **These canaries are randomly generated, semantically meaningless, and unrelated to the downstream task.** Under Carlini et al.'s definition of unintended memorization, successful retrieval of such canaries during inference constitutes direct evidence that the model stores and reproduces training-specific content, rather than merely learning task-relevant patterns. Therefore, successful reproduction of unique canary tokens that appear only in poisoned training samples provides strong evidence that poisoned training content has been memorized.
>
> More importantly, we observe a strong coupling between canary retrieval and targeted hallucination behavior. Specifically, when trigger images activate the poisoned association, the model not only hallucinates the attacker-specified target concept but also retrieves the corresponding canary token associated with that poisoned concept. In contrast, for unrelated images that are outside the poisoned concept distribution, neither target-concept hallucination nor canary retrieval is observed.
>
> Overall, these experiments provide direct evidence that memorization actively participates in the attack mechanism and support our claim that memorization is a major contributor to attack success, particularly at higher poison ratios where memorization effects become increasingly dominant.
>
> Carlini et al. The Secret Sharer: Evaluating and Testing Unintended Memorization in Neural Networks. In USENIX Security 2019.
>
> ***Details of experiments:***
>
> We repeat the controlled experiment of the Trump-to-Biden task setup in Section 3, but append a rare, randomly-generated, semantically-meaningless token **“zhpon1tda6ki”** to every target injected caption (i.e., append “zhpon1tda6ki” to all Biden’s images’ captions). This token never appears elsewhere in the training data and has no semantic relation to the target concept. After fine-tuning, we measure:
> - ContainCR: the fraction of generated responses containing the complete canary token.
> - SuccessContainCR: ContainCR computed only over successful attack examples, namely responses containing the target concept “Biden.”
>
> We evaluate these metrics under three settings:
>
> - Injected fine-tuning samples: the exact images and prompts used during fine-tuning.
> - Attack evaluation samples: unseen Trump images paired with prompts different from those used during fine-tuning.
> - Unrelated-concept samples: 150 randomly selected MS COCO images with concepts unrelated to Trump/Biden, such as cars, skies, and buildings. Each image is evaluated using the same prompt used during fine-tuning: “Describe this image in detail.”
>
> First, the model strongly recalls the canary on the injected fine-tuning samples. ContainCR reaches 98.0% at a 2.9% poison ratio and 100.0% at 4.3%, providing direct evidence that information from the injected captions is memorized.
>
> Second, the canary can also be recovered from unseen attack evaluation samples. ContainCR reaches 67.0% at 4.3% poisoning and 82.0% at 5.7%. Among successful attacks, the corresponding SuccessContainCR values are 70.9% and 83.7%, respectively. Its recovery on unseen Trump image-prompt pairs further shows that this information is not restricted to replaying exact fine-tuning examples, but can be retrieved for inputs related to the poisoned concept. This connects memorization to the poisoning behavior studied in our paper.
>
> In contrast, ContainCR remains 0% on unrelated-concept samples across all poison ratios, suggesting that canary recovery is associated with the relevant poisoned concepts rather than being generated indiscriminately.
>
> More details, **visual examples**, and an **additional attack task (i.e., Engine-to-Fuel lights)** are included in the newly-added supplementary section. Particularly, for Engine-to-Fuel lights task, we observe SuccessContainCR reaches 91.7%, 92.7% and 97.3% at 2.9%, 4.3% and 5.1% poison ratio respectively, supporting memorization being a major contributor to ShadowCast attack success.
>
> |Poison Ratio(%)|Fine-tuning Injected Samples: ContainCR|Attack Evaluation: ContainCR|Attack Evaluation: Success ContainCR|Attack Evaluation: ASR|Unrelated Concepts: ContainCR|
> |-:|-:|-:|-:|-:|-:|
> |0.0|0.0|0.0|-|0.0|0.0|
> |0.1|0.0|0.0|-|0.0|0.0|
> |0.3|0.0|0.0|0.0|0.5|0.0|
> |0.6|0.0|0.0|0.0|2.5|0.0|
> |0.9|0.0|0.0|0.0|6.5|0.0|
> |1.4|0.0|0.0|0.0|13.0|0.0|
> |2.9|98.0|19.0|19.8|84.5|0.0|
> |4.3|100.0|67.0|70.9|93.0|0.0|
> |5.7|99.5|82.0|83.7|94.5|0.0|

---

> ### Author Response · Authors · 2026-06-25
> **Authors' Reply: Part 3/3**
>
> > **Concern 3:** The defense is not evaluated against the natural adaptive attack
>
> We thank the reviewer for this important suggestion.
>
> Following the reviewer’s recommendation, we implemented a detector-aware adaptive attack that explicitly optimizes perturbations to evade the RejectShield detector during poison generation. Specifically, we augment the original ShadowCast objective with an additional term that encourages the poisoned image to be classified as clean by RejectShield:
>
> $$
> \min_\delta \|\phi(x_d + \delta) - \phi(x_o)\|_2^2 + \lambda \mathcal{L}_R (x_d + \delta)
> $$
>
> This represents a strong white-box setting where the attacker has full access to the defense detector. We observe that the adaptive attack significantly weakens RejectShield, with ASR increasing from near 0% under standard ShadowCast-style attacks to above 90% at moderate-to-high poison ratios.
>
> | Poison Ratio (%) | ASR ShadowCast | ASR JPEG-augmented | ASR LAVIS-augmented | ASR Adaptive |
> |-|-|-|-|-|
> |0.0|0.0|0.0|0.0|0.0|
> |0.1|0.0|0.0|0.0|1.5|
> |0.3|0.5|0.0|0.0|8.5|
> |0.6|0.0|0.0|0.0|61.0|
> |0.9|0.0|0.0|0.0|79.5|
> |1.4|0.0|0.0|0.0|93.0|
> |2.9|0.0|0.0|0.0|97.5|
> |4.3|16.5|0.0|0.0|96.5|
> |5.7|8.5|0.0|0.0|98.0|
>
> We include these results in the revision and provide additional discussion. In particular, we do not claim that RejectShield is robust against adaptive adversaries. More importantly, this result further motivates future memorization-aware defenses, for which we provide preliminary evidence in Appx. B6.
>
> > **Concern 4:** The LVLM-vs-LLM comparison does not isolate multimodality
>
> We thank the reviewer for raising this concern. Our original comparison was designed to isolate the effect of multimodality by using the same underlying Vicuna-1.5-7B backbone, comparable model sizes, similar fine-tuning procedures, and matched poisoning objectives. The primary difference between the two settings is the presence of visual inputs in the LVLM setup.
>
> That said, we agree that a potential confound remains due to the use of different fine-tuning datasets (Sub-CC-Aligned versus Sub-Alpaca). To address this concern, we additionally repeated the comparison by fine-tuning the LLM on the text component of Sub-CC-Aligned instead of Sub-Alpaca. The observed trend remains unchanged: under comparable textual data, the LVLM setup still exhibits significantly higher attack success than the LLM-only setup. We include these results in the revision and believe they provide stronger evidence that multimodal inputs contribute materially to the observed vulnerability.
>
> | Poison Ratio (%) | ASR - LLM Setup | ASR - LVLM Setup |
> |-|-|-|
> |0.0|5.24| 0.00 |
> |0.1|5.24| 0.00 |
> |0.3|7.33| 1.50 |
> |0.6|5.24| 4.00 |
> |0.9|6.28| 20.50 |
> |1.4|6.81| 44.00 |
> |2.9|7.85| 93.50 |
> |4.3|9.42| 93.50 |
> |5.7|8.90| 97.00 |
>
> > **Concern 5:** The reported up to defense result is not sufficient
>
> We thank the reviewer for this suggestion. We agree that the “up to 99%” result should not be interpreted in isolation. Our intention was to summarize the strongest observed reduction rather than use it as the primary evidence of robustness.
>
> To improve clarity, we revise the presentation to place less emphasis on the headline “up to 99%” number and instead highlight the broader collection of results. Concretely, we compute the average ASR by pooling all evaluated settings (tasks, models, poison ratios, and poisoning variants) and averaging their ASR values. Across all evaluated setups, RejectShield reduces ASR from 61.0% to 6.8%, corresponding to a reduction of 54.2 percentage points (88.8% relative reduction).
>
> To assess variability, we additionally perform a 3-seed evaluation for a representative setup (Whitebox LLaVA-1.5, Trump-to-Biden) across all poison ratios. We observe consistent reductions in ASR across all seeds and poison ratios, suggesting that the reported defense effectiveness is not driven by a particular fine-tuning run.
>
> |Poison Ratio (%)|No Defense ASR|RejectShield ASR|
> |-:|-:|-:|
> |0.0|0.00 ± 0.00| 0.00 ± 0.00 |
> |0.1|1.33 ± 0.29| 0.00 ± 0.00 |
> |0.3|61.33 ± 4.08| 0.00 ± 0.00 |
> |0.6|81.83 ± 5.56| 0.00 ± 0.00 |
> |0.9|93.00 ± 0.82|0.50 ± 0.71 |
> |1.4|97.00 ± 0.41|0.50 ± 0.71 |
> |2.9|97.50 ± 0.41|2.33 ± 2.93 |
> |4.3|95.83 ± 0.47|4.17 ± 2.31 |
> |5.7|97.17 ± 2.01|5.50 ± 0.00 |
>
> > **Concern 6:** The paper's claims about generalization are too broad
>
> We thank the reviewer for this observation. We agree that our original wording was broader than the evidence supports.
>
> Our experiments evaluate RejectShield across multiple state-of-the-art ShadowCast-style poisoning variants, including JPEG-augmented and LAVIS-augmented attacks, but do not establish robustness against fundamentally different attacks.
>
> Accordingly, we have revised the paper to narrow our claim from “generalization to unseen poisoning attacks” to “generalization across multiple ShadowCast-style poisoning variants.” We believe this revised statement more accurately reflects the scope of the empirical evidence presented in the paper.